# Target Conditioned Representation Independence (TCRI); From Domain-Invariant to Domain-General Representations

## Abstract

We propose a Target Conditioned Representation Independence (TCRI) objective for domain generalization. TCRI addresses the limitations of existing domain generalization methods due to incomplete constraints. Specifically, TCRI implements regularizers motivated by conditional independence constraints that are sufficient to strictly learn complete sets of invariant mechanisms, which we show are necessary and sufficient for domain generalization. Empirically, we show that TCRI is effective on both synthetic and real-world data. TCRI is competitive with baselines in average accuracy while outperforming them in worst-domain accuracy, indicating desired cross-domain stability.

## 1 Introduction

Machine learning algorithms are evaluated by their ability to generalize (generate reasonable predictions for unseen examples). Often, learning frameworks are designed to exploit some shared structure between training data and the expected data at deployment. A common assumption is that the training and testing examples are drawn independently and from the same distribution (iid). Given the iid assumption, Empirical Risk Minimization (ERM; Vapnik (1991)) and its variants give strong generalization guarantees and are effective in practice.

Nevertheless, many practical problems contain distribution shifts between train and test domains, and ERM can fail under this setting (Arjovsky et al., 2019). This failure mode has impactful real-world implications. For example, in safety-critical settings such as autonomous driving (Amodei et al., 2016; Filos et al., 2020), where a lack of robustness to distribution shift can lead to human casualties; or in ethical settings such as healthcare, where distribution shifts can lead to biases that adversely affect subgroups of the population (Singh et al., 2021). To address this limitation, many works have developed approaches for learning under distribution shift. Among the various strategies to achieve domain generalization, Invariant Causal Predictions (ICP; Peters et al. (2016)) has emerged as popular. ICPs assume that while some aspects of the data distributions may vary across domains, the causal structure (or data-generating mechanisms) remains the same and try to learn those domain-general causal predictors.

Following ICP, Arjovsky et al. (2019) propose Invariant Risk Minimization (IRM) to identify invariant mechanisms by learning a representation of the observed features that yields a shared optimal linear predictor across domains. However, recent work (Rosenfeld et al., 2020), has shown that the IRM objective does not necessarily strictly identify the causal predictors, i.e., the representation learn may include noncausal features. Thus, we investigate the conditions necessary to learn the desired domain-general predictor and diagnose that the common domain-invariance Directed Acyclic Graph (DAG) constraint is insufficient to (i) strictly and (ii) wholly identify the set of causal mechanisms from observed domains. This insight motivates us to specify appropriate conditions to learn domain-general models which we propose to implement using regularizers.

**Contributions.** We show that neither a strict subset nor superset of existing invariant causal mechanisms is sufficient to learn domain-general predictors. Unlike previous work, we outline the constraints that identify the strict and complete set of causal mechanisms to achieve domain generality. We then propose regularizers to implement these constraints and empirically show the efficacy of our

proposed algorithm compared to the state-of-the-art on synthetic and real-world data. To this end, we observe that the conditional independence measures are effective for model selection – outperforming standard validation approaches. While our contributions are focused on methodology, our results also highlight existing gaps in standard evaluation using domain-average metrics – which we show can hide worst-case performance; arguably a more meaningful measure of domain generality.

## 2 RELATED WORK

Domain adaptation and generalization have grown to be large sub-fields in recent years. Thus, we do not attempt an exhaustive review, and will only highlight a few papers most related to our work. To address covariate shift, Ben-David et al. (2009) gives bounds on target error based on the $\mathcal{H}$-divergence between the source and target covariate distributions, which motivates domain alignment methods like the Domain Adversarial Neural Networks (Ganin et al., 2016). Others have followed up on this work with other notions of covariant distance for domain adaptation such as mean maximum discrepancy (MMD) (Long et al., 2016) and Wasserstein distance (Courty et al., 2017), etc. However, Kpotufe and Martinet (2018) show that these divergence metrics fail to capture many important properties of transferability, such as asymmetry and non-overlapping support. Zhao et al. (2019) show that even with distribution alignment of covariates, large distances between label distributions inhibit transfer; they propose a label conditional importance weighting method to address this limitation. Additionally, Schrouff et al. (2022) show that many real-world problems contain more complicated 'compound' shifts than covariate shifts. Furthermore, domain alignment methods are useful when one has unlabeled or partially labeled samples from the domain one would like to adapt to during training, however, the domain generalization problem setting may not include such information. The notion of invariant representations starts to address the problem of domain generalization, the topic of this work, rather than domain adaptation.

Arjovsky et al. (2019) propose an objective to learn a representation of the observed features which, when conditioned on, yields a distribution on targets that is domain-invariant, that is, conditionally independent of domain. They argue that satisfying this invariance gives a feature representation that only uses domain-general information. However, Rosenfeld et al. (2020) shows that the IRM objective can fail to recover a predictor that does not use spurious correlations without observing a number of domains greater than the number of spurious features, which can inhibit generalization. Variants of this work Krueger et al. (2021); Wang et al. (2022) address this problem by imposing higher order moment constraints to reduce the necessary number of observed domains. However, as we will show, invariance on the observed domain is not sufficient for domain generalization.

Additionally, one of the motivations for domain generalization is mitigating the worst domain performance. Gulrajani and Lopez-Paz (2020) observe empirically that ERM is competitive and often best in worst domain accuracy across a range of real-world datasets. Rosenfeld et al. (2022) analyze the task of domain generalization as extrapolation via bounded affine transformations and find that ERM remains minimax optimal in the linear regime. However, the extent of the worst-domain shift is often unknown in practice and may not be captured by bounded affine transformations Shen et al. (2021).

In contrast, our work allows for arbitrary distribution shifts, provided that invariant mechanisms remain unchanged. In addition, we show that our proposed method gives a predictor that recovers all domain-general mechanisms and is free of spurious correlations without necessitating examples (neither labeled nor unlabeled) from the target domain.

## 3 PROBLEM SETUP

We consider the data generating mechanism as described by the causal graph in Figure 1 and the equivalent structural equation model (or structural causal model $\mathcal{SCM}$ (equation 1; Pearl (2010)). One particular setting where this graph applies is medicine where we are often interested in predicting conditions from potential causes and symptoms of the condition. Additionally, these features may be influenced by demographic factors that may vary across hospitals (Schrouff et al., 2022). Additionally, in physical processes where measurement is slow, one observes both upstream (causal) and downstream (anticausal) features of the events of interest. An example is in task-fMRI where the BOLD (Blood-Oxygen-Level-Dependent) signal in task-fMRI (functional Magnetic Resonance

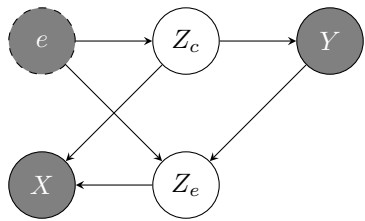

Figure 1: Graphical model depicting the structure of our data-generating process - shaded nodes indicate observed variables. $X$ represents the observed features, $Y$ represents observed targets, and $e$ represents domain influences. There is an explicit separation of domain-general (causal) $Z_c$ and domain-specific (anticausal) $Z_e$ features, which are combined to generate observed $X$.

Imaging) is much slower than the neural processes in the brain associated with the task (Glover, 2011). Many other real-world problems fall under this causal and anticausal setting; we note that this graph is also assumed by previous work Arjovsky et al. (2019). We also assume that the observed data are drawn from a set of $E_{tr}$ training domains $\mathcal{E}_{tr} = \{e_i : i = 1, 2, \ldots, E_{tr}\}$, all generated from Equation 1, thereby fixing the mechanisms by which the observed distribution is generated:

$$\mathcal{SCM}(e_i) \coloneqq \begin{cases} z_c^{(e_i)} \sim P_{Z_c}^{(e_i)} \\ y^{(e_i)} = f_y\left(z_c^{(e_i)}, \nu\right) & \text{where } \nu \perp\!\!\!\perp z_c^{(e_i)}, \\ z_e^{(e_i)} = f_{z_e}\left(y^{(e_i)}, \eta^{(e_i)}\right) & \text{where } \eta^{(e_i)} \perp\!\!\!\perp y^{(e_i)}, \end{cases} \quad (1)$$

where $P_{Z_c}$ is the causal covariate distribution, $f_y$, $f_{z_e}$ are generative mechanisms of $y$ and $z_e$, respectively, and $\nu$, $\eta$ are exogenous variables. These mechanisms are assumed to hold for any domain generated by this generative process, i.e., $\mu_{e_i}(y \,|\, z_c) = \mu(y \,|\, z_c)$ and $\nu_{e_i}(z_e \,|\, y) = \nu(z_e \,|\, y) \,\forall e_i \in \mathcal{E}$ for some distributions $\mu$ and $\nu$, where $\mathcal{E}$ is the set of all possible domains. Under the Markov assumption, we can immediately read off some properties of any distribution induced by the data-generating process shown in Figure 1: (i) $e \perp\!\!\!\perp Y \,|\, Z_c$, (ii) $Z_c \perp\!\!\!\perp Z_e \,|\, Y, e$, and (iii) $Y \not\perp\!\!\!\perp e \,|\, X$. However, as shown in Figure 1, we only observe an unknown function of latent variables $z_c$ and $z_e$, $x = h(z_c, z_e)$ and we would like predictions that do not depend on domains ($e$). Ideally, we want to map $x \to z_c$ such that the mechanism $f_y$ can be learned, as **this would suffice for domain generalization.**

In contrast, though we have that the mechanism $z_e \to y$ is preserved, we have no such guarantee on the inverse $z_e \not\to y$ as it may not exist or be unique and, *therefore, does not satisfy domain generalization.* This becomes a problem when considering mappings that include $z_e$:

$$\mu_{e_i}(y \,|\, z_e) \neq \mu_{e_j}(y \,|\, z_e) \text{ and}$$
$$\mu_{e_i}(y \,|\, z_c, z_e) \neq \mu_{e_j}(y \,|\, z_c, z_e) \text{ for } i \neq j.$$

The latter implies that $\mu_{e_i}(y \,|\, x) \neq \mu_{e_j}(y \,|\, x)$ for $i \neq j$, and therefore, the original observed features will not be domain-general. Note that the generative process and its implicates are vital for our approach, as we assume that it is preserved across domains.

**Assumption 3.1.** We assume that all distributions, train, and test (observed and unobserved at train), are generated by the generative process described in Equation 1.

In the following sections, we will introduce our proposed algorithm to learn a feature extractor that maps to $Z_c$ and show the generative properties that are necessary and sufficient to do so.

## 4 TARGET-CONDITIONED REPRESENTATION INDEPENDENCE OBJECTIVE

We first define two distinct types of representations of observed features that we will need henceforth – domain-invariant and domain-general.

**Definition 4.1.** A **domain-invariant** representation $\Phi(X)$, with respect to a set of observed domains $\mathcal{E}_{tr}$, is one that satisfies the intervention properties $\mu_{e_i}(y|\Phi(X) = x) = \mu_{e_j}(y|\Phi(X) = x) \,\forall e_i, e_j \in \mathcal{E}_{tr}$ for any fixed $x$, where $e_i$, $e_j$ are domain identifiers and $\mu$ is a probability distribution.

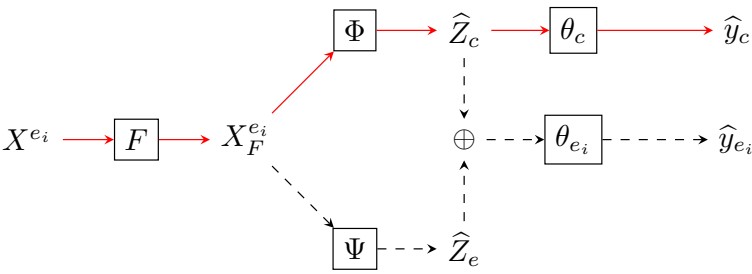

Figure 2: We first generate a feature representation via featurizer $F$. During training, both representations, $\Phi$ and $\Psi$, generate predictions – domain-general and domain-specific predictions, respectively. However, only the domain-invariant representations/predictions are used during test time – indicated by the solid red arrows. $\oplus$ indicates concatenation.

In other words, under a domain-invariant representation, the output conditional distribution for a given input is necessarily the same across the reference (typically observed) domains. This is consistent with the existence of $\mu$ s.t. $\mu\left(y|\Phi(X)=x\right) = \mu\left(y|\mathrm{do}(\Phi(X)=x)\right)$, where $\mathrm{do}(\cdot)$ is the "do-operator" Pearl (2010), denoting an intervention (arbitrary assignment of value). Conversely, for a domain-specific representation, the output conditional distribution for a given input need not be the same across domains.

However, by definition, this representation is domain-invariant up to a specific set of domains, often the set of domains that it is learned on. So, it may not be domain-invariant with respect to test domains without additional assumptions on the set of training domains, e.g., their convex hull (Rosenfeld et al., 2020). So, we need to refine this property to better specify the representations we would like to learn. This motivates the following definition which ties domain-invariance to a specific generative process, as opposed to a set of the observed distributions, which, along with Assumption 3.1, connects causality to domain-generality.

**Definition 4.2.** A representation is **domain-general** with respect to a DAG $\mathcal{G}$ if it is domain-invariant with respect to $\mathcal{E}$, where $\mathcal{E}$ is the set of all possible domains generated by $\mathcal{G}$.

By Assumption 3.1, causal mechanisms from features $Z_c$ to target $Y$ are domain-general, so a natural strategy is to extract $Z_c$ from the observed features. In order to understand when we can recover $Z_c$ from the observed $X$, we will consider two conditional independence properties implied by the assumed causal graph: $Y \perp\!\!\!\perp e \mid Z_c$, which we call the *domain invariance property*, and $Z_c \perp\!\!\!\perp Z_e \mid Y, e$, which we capture in the following *target conditioned representation independence property* (TCRI; definition 4.3).

**Definition 4.3** (Target Conditioned Representation Independence)**.** Two functions, $\Phi$, $\Psi$, are said to satisfy TCRI with respect to random variables $X, Y, e_i$ if $I(\Phi(X), \Psi(X); Y) = I(Z_c, Z_e; Y)$ (total-chain-information-criterion) and $\Phi(X) \perp\!\!\!\perp \Psi(X) \mid Y \,\forall e_i$.

We will show that these properties together identify $Z_c$ from $X, Y$ to give a domain-general representation in Section 5. Based on our results, we design an algorithm to learn a feature mapping that recovers $Z_c$, i.e., the domain-general representation (mechanisms). Figure 2 illustrates the learning framework.

In practice, we propose a TCRI objective containing four terms, each related to the properties desired of the learned representations, as follows,

$$\mathcal{L}_{TCRI} = \frac{1}{E_{tr}} \sum_{e_i \in \mathcal{E}_{tr}} \left[ \alpha \mathcal{L}_{\Phi} + (1-\alpha)\mathcal{L}_{\Phi\oplus\Psi} + \lambda \mathcal{L}_{IRMv1'} + \beta \mathcal{L}_{CI} \right], \quad (2)$$

where $\alpha \in [0, 1]$, $\beta > 0$, and $\lambda > 0$ are hyperparameters – Figure 2 shows the full framework.

In detail, we let $\mathcal{L}_{\Phi} = \mathcal{R}(\theta_c \cdot \Phi)$ represent the domain-general predictor accuracy and let $\mathcal{L}_{IRMv1'}$ be a penalty from on $\Phi$ and the linear predictor $\theta_c$ that enforces that $\Phi$ has the same optimal predictor across training domains, capturing the domain invariance property, where $\Phi : \mathcal{X} \mapsto \mathcal{H}_{\Phi}$, $\theta_c : \mathcal{H}_{\Phi} \mapsto \mathcal{Y}$ (Arjovsky et al., 2019):

$$\mathcal{L}_{IRMv1'} = \|\nabla_{\theta_c} \mathcal{R}^{e_i}(\theta_c \circ \Phi)\|_{2},$$

where $\mathcal{R}^{e_i}$ denotes the empirical risk achieved on domain $e_i$. The $\mathcal{L}_\Phi$ and $\mathcal{L}_{IRMv1'}$ together implement a domain-invariance property, however, we know that this is not sufficient for domain generalization (Rosenfeld et al., 2020). We will show later that the addition of the TCRI property suffices for domain generalization (Theorem 5.4).

To implement the TCRI property (definition 4.3), we also learn a domain-specific representation $\Psi$, which is constrained to be (i) conditionally independent of the domain-general representation given the target and domain and (ii) yield a predictor as good as one from $X$ when combined with $\Phi$. We first address (ii); given a domain-specific representation, $\Psi : \mathcal{X} \mapsto \mathcal{H}_\Psi$, we define a set of domain-specific predictors $\{\theta_{e_i} : \mathcal{H}_\Phi \times \mathcal{H}_\Psi \mapsto \mathcal{Y} : i = 1, \ldots, E_{tr}\}$. We then add a term in the objective that minimizes the loss of these domain-specific predictors:

$$\mathcal{L}_{\Phi \oplus \Psi} = \mathcal{R}^{e_i} \left( \theta_{e_i} \circ (\Phi \oplus \Psi) \right).$$

This term aims to enforce the total-chain-information-criterion of TCRI, by allowing $\Phi$ and $\Psi$ to minimize a domain-specific loss together, where non-domain-general information about the target can be used to improve performance. Since we have that both the domain-general and domain-specific have unique information about the target, the optimal model will use both types of information. We allow the $\theta_{e_i}$ for each domain to be different since, by definition of the problem, we expect these mechanisms to vary across domains.

We define $\mathcal{L}_{CI}$ to be the conditional independence part of the TCRI property and use the V-statistic-based Hilbert-Schmidt Independence Criterion (HSIC) estimate (more details on HSIC can be found in Gretton et al. (2007)). For the two representations $\Phi(X)$, $\Psi(X)$ for which we want to determine independence ($\Phi(X) \perp\!\!\!\perp \Psi(X)$), define

$$\mathcal{L}_{CI} = \frac{1}{C} \sum_{k=1}^{C} \widehat{HSIC}(\Phi(X)_k, \Psi(X)_k) = \frac{1}{C} \sum_{k=1}^{C} \frac{1}{n_k^2} \text{trace}(\mathbf{K}_\Phi \mathbf{H}_{n_k} \mathbf{K}_\Psi \mathbf{H}_{n_k}),$$

where $k$, indicates which class the examples in the estimate correspond to, $C$ is the number of classes, $\mathbf{K}_\Phi \in \mathbb{R}^{n_k \times n_k}$, $\mathbf{K}_\Psi \in \mathbb{R}^{n_k \times n_k}$ are Gram matrices, $\mathbf{K}_\Phi^{i,j} = \kappa(\Phi(X)_i, \Phi(X)_j)$, $\mathbf{K}_\Psi^{i,j} = \omega(\Psi(X)_i, \Psi(X)_j)$ with kernels $\kappa, \omega$ are radial basis functions, $\mathbf{H}_{n_k} = \mathbf{I}_{n_k} - \frac{1}{n_k^2} \mathbf{1}\mathbf{1}^\top$ is a centering matrix, $\mathbf{I}_{n_k}$ is the $n_k \times n_k$ dimensional identity matrix, $\mathbf{1}_{n_k}$ is the $n_k$-dimensional vector whose elements are all 1, and $^\top$ denotes the transpose. We condition on a label and domain by taking only examples of each label-domain pair and computing the empirical HSIC; then we take the average across all labels. We note that any criterion for conditional independence can be used as $\mathcal{L}_{CI}$, e.g., partial covariance.

Altogether, we have the following task:

$$\min_{\Phi, \Psi, \theta_c, \theta_1, \theta_2, \ldots, \theta_{E_{tr}}} \frac{1}{E_{tr}} \sum_{e_i \in \mathcal{E}_{tr}} \left[ \alpha \mathcal{L}_\Phi + (1 - \alpha)\mathcal{L}_{\Phi \oplus \Psi} + \mathcal{L}_{IRMv1'} + \beta \mathcal{L}_{CI} \right]. \tag{3}$$

We compute the complete objective for each domain separately to condition on the domains $e$, and after minimizing this objective, only the invariant representation and its predictor, $\theta_c \circ \Phi$, are used.

## 5 Conditions for a Domain-General Representations

Now we provide some analysis to justify our method. Consider a feature extractor $\Phi : \mathcal{X} \mapsto \mathcal{Z}$, where $\mathcal{X}, \mathcal{Z}$ are input and latent features spaces, respectively. We first show that a $\Phi$ that captures a strict subset of the causal features satisfies the domain invariance property $Y \perp\!\!\!\perp e \,|\, \Phi(X)$ while not necessarily being domain-general.

**Lemma 5.1** (Insufficiency of Causal Subsets for domain generalization). *Conditioning on a subset of causal variables (invariant mechanisms) does not imply domain generalization (definition 4.2).*

$$\mathcal{Z} \subset \mathcal{Z}_c \;\not\!\!\implies\; \mu_{e_i}(y \,|\, Z = z) = \mu_{e_j}(y \,|\, Z = z) \forall e_i \neq e_j, \, z \in \mathcal{Z}$$

*where $\mathcal{Z}_c$ is the causal feature space.*

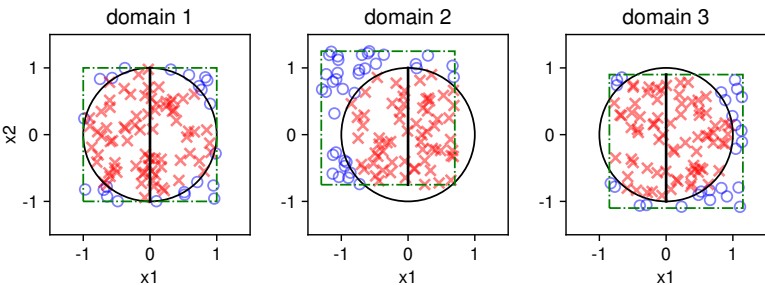

Figure 3: Visualization of Lemma 5.1, where $x_1 = z_1$ and $x_2 = z_2$. The large circle is the true decision boundary, where $\times : y = 0, \circ : y = 1$. The dotted square indicates the covariate distribution (Uniform), which differs across domains. The length of the solid line indicates the $\mu(y = 0 \,|\, z_1 = 0)$. Clearly the length of each line changes across domains, hence the causal subset $z_1$ does not yield a domain-general predictor.

*Proof.* We provide a simple counterexample. Suppose we have the following generative process $d_{e_i,1} \to z_1 \to y \leftarrow z_2 \leftarrow d_{e_i,2}$ with $z_1, z_2 \in \mathbb{R}$, $d_{e_i,1}, d_{e_i,2} \in \mathbb{R}_{\geq 0}$, $r \in \mathbb{R}_{>0}$, and

$$y = \begin{cases} 0 \text{ if } z_1^2 + z_2^2 <= r \\ 1 \text{ if } z_1^2 + z_2^2 > r. \end{cases}$$

Suppose we observe domains $\{e_i : i = 1, \ldots, E_{tr}\}$, where $z_1 \sim \text{Uniform}(-r + d_{e_i,1}, r + d_{e_i,1})$ and $z_2 \sim \text{Uniform}(-r + d_{e_i,2}, r + d_{e_i,2})$ for domain $e_i$ and $d_{e_i}$ is a domain-specific quantity. Now, suppose we condition on $z_1$. The optimal predictor is

$$\mu_{e_i}(y = 1 \,|\, z_1) = \frac{2\sqrt{r^2 - z_1^2} - d_{e_i,2}}{2r}.$$

In this case, $z_1$, a causal subset, does not yield domain-general representation since its optimal predictor depends on $d_{e_i}$, which changes across domains. $\square$

We provide a visualization of Lemma 5.1 in Figure 3. Additionally, we show that subsets of the causal features can satisfy the domain-invariance property.

**Lemma 5.2.** *A representation $\Phi$ that maps to a strictly causal subset can be Domain-invariant (definition 4.1).*

*Proof.* This proof follows a similar argument as Lemma 5.1. We replace the mechanism for $z_1$ with $z_1 \sim \text{Uniform}(d_{e_i,1} \cdot -r, d_{e_i,1} \cdot r)$, i.e., the domain is scaled along $z_1$. Suppose $d_{e_i,2} = 0$ for all training domains $\{e_i : i = 1, \ldots, E_{tr}\}$ and $\Phi(z_1, z_2) = z_1$. There is still a covariate distribution shift due to varying values (scaling) of $d_{e_i,1}$. Then $\Phi$ is domain invariant on the training domain but will no longer be domain-general with respect to any domain $e_j$ where $d_{e_j,2} > 0$. $\square$

In other words, one failure case that illustrates a limitation of strictly considering the domain-invariance property ($\Phi(X) \perp\!\!\!\perp Y \,|\, e$) is when the shifts observed in the training set are different than those in the test set. Specifically, an incomplete causal representation may not be domain-general if the shifts at test-time are on causal features that are not captured in the incomplete causal representation. We now show that the domain-invariance property and Target Conditioned Representation Independence (TCRI) property are sufficient for recovering the complete set of invariant mechanisms.

**Lemma 5.3.** *(Sufficiency of TCRI for Causal Aggregation). Recall, $X, Z_c, Z_e, Y$ from Figure 1. Let $Z_c, Z_e$ be direct causes and direct effects of $Y$, respectively, and recall that $X$ is a function of $Z_c$ and $Z_e$. If the two representations induced by feature extractors $\Phi, \Psi$ satisfy TCRI, then wlog $I(\Phi(X); Y) \geq I(Z_c; Y)$.*

Lemma 5.3 shows that the TCRI property directly solves the problem identified in Lemma 5.1, that is, the TCRI property implies that all causal information about the target is aggregated into one representation $\Phi$. Now, to identify $Z_c$, we only need to show that $\Phi$ is strictly causal.

**Theorem 5.4** (Sufficiency of TCRI + domain-invariance properties for identifying $Z_c$). *Recall that $Z_c, Z_e$ are the true latent features in the assumed generative model and $X$ are the observed features – Figure 1. If $\Phi$, $\Psi$ satisfy TCRI and domain-invariance property, then $\Phi$ recovers $Z_c$ and is therefore domain-general.*

*Proof.* By Lemma 5.3, we have that when $\Phi$, $\Psi$ satisfy TCRI, $\Phi(X)$ contains all of the causal (necessarily domain-general) information in $X$. However, $\Phi(X)$ may also contain non-causal (domain-specific) information, spoiling the domain-generality of $\Phi$. It remains to show that $\Phi(X)$ is strictly causal when we add the domain-invariance property.

If $\Phi$ satisfies the domain-invariance property, then $Y \perp\!\!\!\perp e \,|\, \Phi(X)$. Clearly, this cannot be the case if $\Phi(X)$ contains features of $Z_e$, since $e$ and $Y$ are colliders on $Z_e$ and therefore conditioning on $Z_e$ opens a path between $e$ and $Y$, making them dependent. Thus $\Phi(X)$ can only contain causal features.

Therefore a representation that satisfies TCRI and the domain-invariance property is wholly and strictly causal and thus domain-general. The latter follows from the $Z_c$ having invariant mechanisms. $\square$

Theorem 5.4 suggests that by learning two representations that together capture the mutual information between the observed $X$, $Y$, where one satisfies the domain-invariance property and both satisfy TCRI, one can recover the strictly and complete causal feature extractor and domain-general predictor.

*Remark* 5.5. (Revisiting Lemma 5.1's counterexample) Given two representations $\Phi$, $\Psi$ that satisfy TCRI, $\Phi$ necessarily captures $z_1, z_2$. By definition, $\Phi$, $\Psi$ must capture all of the information in $z_1, z_2$ about $y$, and we know from the graph that they are conditionally dependent given $y$, i.e, $z_1, z_2$ are common causes of $y$ (colliders), so conditioning on $y$ renders the marginally independent variables dependent. So, $z_1$ and $z_2$ must be captured the same feature extractor.

*Remark* 5.6. One limitation of TCRI is a failure mode when the strictly anticausal representation gives a domain invariant predictor. In this case, either representation may be $\Phi$. However, one of the benefits of having a domain-specific predictor for each observed domain is that we can check if those classifiers are interchangeable. Specifically, if we are in this scenario where the causal features are mapped to the domain-specific feature extractor, we will see that the domain-specific classifiers give similar results when applied to a domain that they were not trained on since they are based on invariant causal mechanisms. This, however, gives a test not a fix for this setting – we leave a fix for future work.

## 6 EXPERIMENTS

$$\mathcal{SCM}(e_i) := \begin{cases} z_c^{(e_i)} \sim \mathrm{Exp}\left(\sigma^{e_i}\right) \\ y^{(e_i)} = z_c^{(e_i)} + \mathrm{Exp}\left(0.25\right), \\ z_e^{(e_i)} = y^{(e_i)} + \mathrm{Exp}\left(\sigma_\eta^{e_i}\right). \end{cases} \quad (4)$$

| Model | $\Phi_{0,0}$ | $\Phi_{1,0}$ |
|---|---|---|
| ERM | 0.84 | 0.18 |
| IRM | 0.83 | 0.18 |
| TCRI (HSIC) | 1.11 | 0.01 |
| Oracle | 1.13 | 0.0 |

Table 1: Continuous Simulated Results – Feature Extractor with a dummy predictor $\theta_c = 1.$, i.e., $\hat{y} = x \times \Phi \times w$, where $x \in \mathbb{R}^{N \times 2}$, $\Phi \in \mathbb{R}^{2 \times 1}$, $w \in \mathbb{R}$. Oracle indicates the coefficients achieved by regressing $y$ on $z_c$ directly.

To evaluate our method in a setting that exactly matches our assumptions and we know the ground truth mechanisms, we use Equation 4 to generate our linear-Gaussian data, with domain parameters $\sigma^{e_i}$, $\sigma_\eta^{e_i}$ – code provided in the supplemental materials.

We observe 2 domains with parameters $\sigma^{e_i=0} = \sigma_\eta^{e_i=0} = 0.1$, $\sigma^{e_i=1} = \sigma_\eta^{e_i=1} = 1$, each with 1000 samples, and use linear feature extractors and predictors. Minimizing the TCRI objective (Equation 3) recovers a linear feature representation that maps back to $z_c$ (Table 1). Note that for

Table 2: Colored MNIST. 'acc' indicates model selection via validation accuracy, and 'cov' indicates model selection via validation conditional independence. 'ci' indicates a conditional cross-covariance penalty, and 'HSIC' indicates a Hilbert-Schmidt Independence Criterion penalty. Columns {+90%, +80%, -90%} indicate domains – {0.1, 0.2, 0.9} digit label and color correlation, respectively. $\hat{\mu}$, $\hat{\sigma}$ indicate the mean and standard deviation of the average domain accuracies, over 3 trials each, respectively.

| | Domains | | | Domain Accuracy Statistics | | |
|---|---|---|---|---|---|---|
| **Algorithm** | **+90%** | **+80%** | **-90%** | **mean** | **std** | **min** |
| ERM | $71.8 \pm 0.1$ | $72.8 \pm 0.2$ | $10.0 \pm 0.1$ | 51.5 | 36.0 | 10.0 |
| IRM | $72.4 \pm 0.5$ | $72.8 \pm 0.2$ | $10.0 \pm 0.3$ | 51.7 | 36.1 | 10.0 |
| VREx | $72.3 \pm 0.4$ | $73.2 \pm 0.5$ | $10.0 \pm 0.1$ | 51.8 | 36.2 | 10.0 |
| GroupDRO | $72.4 \pm 0.2$ | $73.1 \pm 0.2$ | $10.0 \pm 0.2$ | 51.8 | 36.2 | 10.0 |
| MLDG | $71.7 \pm 0.0$ | $73.0 \pm 0.1$ | $10.2 \pm 0.0$ | 51.6 | 35.9 | 10.2 |
| ARM | $81.9 \pm 0.6$ | $74.5 \pm 1.2$ | $10.2 \pm 0.0$ | 55.6 | 39.4 | 10.2 |
| TCRI (cov) – acc | $70.0 \pm 1.4$ | $70.5 \pm 1.9$ | $10.0 \pm 0.1$ | 50.2 | 34.8 | 10.0 |
| TCRI (HSIC) – acc | $71.9 \pm 0.5$ | $72.6 \pm 0.4$ | $10.3 \pm 0.1$ | 51.6 | 35.8 | 10.3 |
| TCRI (cov) – ci | $54.7 \pm 1.1$ | $56.4 \pm 2.4$ | $50.6 \pm 0.1$ | 53.9 | **3.0** | 50.6 |
| TCRI (HSIC) – ci | $54.7 \pm 1.6$ | $60.1 \pm 4.1$ | $53.0 \pm 2.1$ | **55.9** | 3.6 | **53.0** |

ERM, $\lambda = \beta = 0$, $\alpha = 1$, IRM, $\lambda = 0.1$, $\beta = 0$, $\alpha = 1$, and TCRI, $\lambda = 0.1$, $\beta = 10$ and $\alpha = 0.75$; additional details can be found in Appendix A.

## 6.1 DATASETS

**Algorithms:** We compare our method to the following baselines: **Empirical Risk Minimization:** Empirical Risk Minimization (**ERM**, Vapnik (1991)), Invariant Risk Minimization (**IRM** Arjovsky et al. (2019)), Variance Risk Extrapolation (**V-REx**, Krueger et al. (2021)), Meta-Learning for Domain Generalization (**MLDG**, Li et al. (2018)), Group Distributionally Robust Optimization (**GroupDRO**, Sagawa et al. (2019)), and Adaptive Risk Minimization (**ARM** Zhang et al. (2021)). Additional discussion on the algorithms can be found in the appendix.

We evaluate our proposed method on real-world datasets. Given observed domains $\mathcal{E}_{tr} = \{e_i : i = 1, 2, \ldots, E_{tr}\}$, we train on $\mathcal{E}_{tr} \setminus e_i$ and evaluate the model on the unseen domain $e_i$, for each $e_i$.

**Model Selection:** Typically, ML practitioners use a within-domain hold-out validation set for model selection. However, this strategy is biased towards the empirical risk minimizer, i.e., the one with the lowest error on the validation set from the training domains, however, we know that the model that achieves the highest validation accuracy may not be domain-general. This same is true if we use an out-of-domain validation set that is not from the target domain. Alternatively, we propose to leverage the generative assumptions for model selection. Specifically, we consider other properties of our desired model for model selection; specifically, a low $\mathcal{L}_{CI}$. To do this, we follow the practice of a hold-out within-domain validation set, however, we compute $\mathcal{L}_{CI}$ for the validation set and choose the example with the lowest CI score instead of the highest validation accuracy. We compare this strategy with validation accuracy in our results. Additional details can be found in Appendix C

**ColoredMNIST:** We evaluate our method on the ColoredMNIST dataset Arjovsky et al. (2019) which is composed of 7000 ($2 \times 28 \times 28$, 1) images of a hand-written digit and binary-label pairs. There are three domains with different correlations between image color and label, i.e., the image color is spuriously related to the label by assigning a color to each of the two classes (0: digits 0-4, 1: digits 5-9). The color is then flipped with probabilities {0.1, 0.2, 0.9} to create three domains, making the color-label relationship domain-specific, as it changes across domains. There is also label flip noise of 0.25, so we expect that the best we can do is 75% accuracy. As in Figure 1, $Z_c$ corresponds to the original image, $Z_e$ the color, $e$ the label-color correlation, $Y$ the image label, and $X$ the observed colored image. Code (a variant of https://github.com/facebookresearch/DomainBed) can be found at https://anonymous.4open.science/r/DomainBed-8D3F.

We use MNIST-ConvNet Gulrajani and Lopez-Paz (2020) backbones for the MNIST datasets and parameterize our experiments with the DomainBed hyperparameters with three trials to select the best model Gulrajani and Lopez-Paz (2020). The MNIST-ConvNet backbone corresponds to the generic featurizer $F$ in Figure 2, and both $\Phi$ and $\Psi$ are linear layers of size $128 \times 128$ that are appended to the backbone. The predictors $\theta_c, \theta_1, \ldots, \theta_{E_{tr}}$ are also parameterized to be linear and appended to the $\Phi$ and $\Psi$ layers, respectively.

## 6.2 Results and Discussion

We observe that TCRI is competitive in average accuracy with the baseline methods. However, it is significantly more stable when using conditional independence for model selection, i.e., the worst-domain accuracy is highest and variance across domains is lowest for TCRI – both by a large margin. We note that domain -90%, which has a color-label relationship flip probability of 0.9, has a majority color-label pairing that is opposite of domains +80% and +90%, with flip probabilities of 0.1 and 0.2, respectively. Hence, we observe that the baseline algorithms generalize poorly to domain -90%, relative to the other two domains. This is a clear indication that, unlike in TCRI, the baselines are using spurious information (color) for prediction. While TCRI does not obtain the expected best accuracy of 75%, it is evident that the information being used for prediction is general across the three domains, given the low variance in cross-domain accuracy.

**Worst-domain Accuracy:** An important implication of a domain-general is stability – robustness in worst-domain performance, up to domain difficulty. While average accuracy across domains, provides some insight into an algorithm's ability to generalize to new domains, it is susceptible to being dominated by the performance of subsets of observed domains. For example, ARM outperforms the baselines in average accuracy, but this improvement is driven primarily by the first domain (+90%), while the worst-domain accuracy stays the same. In the context of real-world challenges such as algorithmic fairness, comparable worst-domain accuracy is necessary (Hardt et al., 2016). TCRI achieves 5x (53.0 vs. 10.2) the worst-domain accuracy of the best baseline while maintaining the competitive average accuracy – outperforming most of the baselines on average. Additionally, the variance of the domain accuracies is over 10x lower than that of the baselines, showing further evidence of the cross-domain stability one would expect from a domain-general algorithm. The baselines also include V-REx, which implements regularizers on risk variance across observed domains.

**Limitations:** The strength of TCRI is also its limitation; TCRI is very conservative, so as to be robust to worst-domain shifts. While many important real-world problems require robustness to worst-domain shifts, this is not always the case, and in this setting, TCRI sacrifices performance gains from non-domain-general information that may be domain-invariant with respect to the expected domains. The practitioner should apply this method when it is appropriate, i.e., when domain generality is critical and the target domains may be sufficiently different than the source domain. It is, however, important to note that in many settings where one is happy with domain-invariance as opposed to domain-generality, ERM may be sufficient (Gulrajani and Lopez-Paz, 2020; Rosenfeld et al., 2022).

## 7 Conclusion and Future Work

We address the limitations of state-of-the-art algorithms' inability to learn domain-general predictors by developing an objective that enforces DAG properties that are sufficient to disentangle causal (domain-general) and anticausal mechanisms. We compare domain generality to domain-invariance – showing that our method is competitive with other state-of-the-art domain-generalization algorithms on real-world datasets in terms of average across domains. Moreover, TCRI outperforms all baseline algorithms in worst-domain accuracy, indicating desired stability across domains. We also find that using conditional independence metrics for model selection outperforms the typical validation accuracy strategy. Future work includes further investigating other model selection strategies that preserve the desired domain-generality properties and curating more benchmark real-world datasets that exhibit worst-case behavior.

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

## A  SIMULATED DATA

We observe 2 domain with parameters $\sigma^{e_i=0} = \sigma_\eta^{e_i=0} = 0.1$, $\sigma^{e_i=1} = \sigma_\eta^{e_i=1} = 1$, each with 1000 samples. We let , and use linear feature extractors and predictors. Minimizing the TCRI objective (Equation 3) recovers a linear feature representation that maps back to $z_c$ (Table 1). Note that for ERM, $\lambda = \beta = 0$, $\alpha = 1$, IRM, $\lambda = 0.1$, $\beta = 0$, $\alpha = 1$, and TCRI, $\lambda = 0.1$, $\beta = 10$ and $\alpha = 0.75$; additional details can be found in Appendix A.

In addition to letting $\theta_c = 1$. be a dummy variable, we also solve the OLS (Ordinary Least Squares – $(X^\top X)^{-1} X^\top y$) problem to compute the $\mathcal{L}_{\Phi \oplus \Psi}$ term in the loss. Each backward pass takes in all examples from a domain as a batch.

## B  HYPERPARAMETER SELECTION

We do a random search over hyperparameters for our method – six randomly selected hyperparameter sets in total. Additionally, for each set, we run three trials to generate standard errors. Additional sampling details can be found in https://anonymous.4open.science/r/DomainBed-8D3F/domainbed/hparams_registry.py. We use the default values for the baseline algorithms since the results closely match those reported by Gulrajani and Lopez-Paz (2020). The hyperparameters used for each trial are also provided in the supplemental material.

## C  MODEL SELECTION

Across the hyperparameter sweep, we select the model with the lowest average conditional independence score between the two TCRI representations (Definition 4.3) to evaluate on our test set. This is in lieu of selecting the model with the highest validation accuracy on the training domains.

Additionally, we show results based on oracle selection, that is, selection based on held-out target domain data. We observe that TCRI still outperforms the baseline methods and has accuracies in the source domains that are closer to the baselines. This suggests that the regularizers are not so harsh that the TCRI models are not able to learn good predictors in practice. The results, however, do suggest that there is room for improvement in model selection. We leave this for future work.

Table 3: Colored MNIST. Columns {+90%, +80%, -90%} indicate domains – $\{0.1, 0.2, 0.3, 0.9\}$ digit label and color correlation, respectively. $\hat{\mu}$, $\hat{\sigma}$ indicate the mean and standard deviation of the average domain accuracies, over 3 trials each, respectively. Using the oracle selection method – held out target data.

|  | Domains | | | Domain Accuracy Statistics | | |
|---|---|---|---|---|---|---|
| **Algorithm** | **+90%** | **+80%** | **-90%** | **mean** | **std** | **min** |
| ERM | 61.9 | 66.3 | 26.5 | 51.6 | 17.8 | 26.5 |
| IRM | 73.0 | 72.2 | 51.0 | 65.4 | 10.2 | 51.0 |
| GroupDRO | 64.8 | 68.0 | 26.0 | 53.0 | 19.1 | 26.0 |
| MLDG | 68.8 | 72.7 | 28.6 | 56.7 | 19.9 | 28.6 |
| ARM | **81.6** | 73.4 | 24.2 | 59.7 | 25.3 | 24.2 |
| VREx | 70.2 | 70.8 | 49.7 | 63.6 | 9.8 | 49.7 |
| TCRI (cov) | 61.6 | 66.4 | 53.0 | 60.3 | 5.5 | 53.0 |
| TCRI (HSIC) | 68.2 | 67.7 | **56.4** | 64.1 | **5.4** | **56.4** |

## D  ON BENCHMARK DATASETS FOR EVALUATING DOMAIN GENERALIZATION – WORST-CASE

We show some results below that illustrate the challenge of accurately evaluating the efficacy of an algorithm in domain generalization. We first note that we expect ERM (naive) to perform poorly in domain generalization tasks, certainly so when we observe worst-case shifts at test time. However, like other works (Gulrajani and Lopez-Paz, 2020), we observe that ERM performs as well as other baselines during transfer on various benchmark datasets. Previous theoretical results (Rosenfeld et al.,

2022) suggest that this observation is indicative of properties of the benchmark domains that may be sufficient for domain generalization with ERM - specifically that the distribution (and equivalently loss) of the target domain can be written as a convex combination of the those in the source domains.

To further investigate this, we develop additional experiments motivated by the ColoredMNIST (Arjovsky et al., 2019) which seems to not fall into the scenario in (Rosenfeld et al., 2022). We note that in the +90%, +80%, and -90% domains of ColoredMNIST, the -90% domain has the opposite relationship between the spurious correlation and the label, so the use of spurious correlation generalizes catastrophically in the -90% domain. In the setting, the baseline algorithms we present achieve poor accuracy in the -90% domain while maintaining high accuracy in the +90% and +80% domains. Consequently, we investigate two settings, *setting a*: +90%, +80%, +70%, -90% domains and *setting b*: +90%, +80%, -80%, -90% domains. In setting a, we add another domain with the majority direction in the relationship between spurious correlation and labels, and in setting b, we add another domain with the minority direction.

We use Oracle model selection (held-out target data) to remove the effect of model selection for all methods in the results. We find that in setting a where we add a domain (+70%) that has spurious correlations that do not generalize the -90% domain, we observe worst-case accuracy across baselines is still very different from the median-case 4).

Table 4: Colored MNIST *setting a*. Columns {+90%, +80%, +70%, -90%} indicate domains – {0.1, 0.2, 0.3, 0.9} digit label and color correlation, respectively. $\hat{\mu}$, $\hat{\sigma}$ indicate the mean and standard deviation of the average domain accuracies, over 3 trials each, respectively. Using the oracle selection method – held out target data.

|  | Domains | | | | Domain Accuracy Statistics | | |
|---|---|---|---|---|---|---|---|
| **Algorithm** | **+90%** | **+80%** | **+70%** | **-90%** | **mean** | **std** | **min** |
| ERM | 72.8 ± 0.3 | 74.7 ± 0.3 | 73.3 ± 0.1 | 16.3 ± 1.5 | 59.3 | 24.8 | 16.3 |
| IRM | 49.0 ± 0.1 | 54.2 ± 2.0 | 50.3 ± 0.3 | 43.8 ± 2.8 | 49.3 | 3.7 | 43.8 |
| GroupDRO | 71.0 ± 0.6 | 72.2 ± 0.3 | 70.7 ± 0.9 | 36.4 ± 4.2 | 62.6 | 15.1 | 36.4 |
| MLDG | 72.8 ± 0.9 | 74.8 ± 0.3 | 72.9 ± 0.3 | 13.6 ± 0.7 | 58.5 | 26.0 | 13.6 |
| ARM | 74.7 ± 0.4 | 74.1 ± 0.2 | 73.1 ± 0.4 | 14.0 ± 1.5 | 59.0 | 26.0 | 14.0 |
| VREx | 74.1 ± 1.3 | 72.6 ± 0.5 | 72.1 ± 0.5 | 19.5 ± 5.5 | 59.6 | 23.2 | 19.5 |
| TCRI (cov) | 68.5 ± 4.4 | 66.4 ± 6.5 | 67.8 ± 2.9 | 53.6 ± 2.3 | 64.1 | 6.1 | 53.6 |
| TCRI (HSIC) | 72.1 ± 1.5 | 73.6 ± 0.4 | 72.6 ± 0.4 | 49.9 ± 0.3 | 67.0 | 9.9 | 49.9 |

However, in setting b, where we add a domain (-80%) that has spurious correlations that generalize to the -90% domain, we observe that the worst-case accuracy is much closer than the median-case – single digit standard deviation across domains 5.

Table 5: Colored MNIST *setting b*. Columns {+90%, +80%, -80%, -90%} indicate domains – {0.1, 0.2, 0.8, 0.9} digit label and color correlation, respectively. $\hat{\mu}$, $\hat{\sigma}$ indicate the mean and standard deviation of the average domain accuracies, over 3 trials each, respectively. Using the oracle selection method – held out target data.

|  | Domains | | | | Domain Accuracy Statistics | | |
|---|---|---|---|---|---|---|---|
| **Algorithm** | **+90%** | **+80%** | **-80%** | **-90%** | **mean** | **std** | **min** |
| ERM | 58.4 ± 1.3 | 67.0 ± 0.5 | 64.2 ± 2.0 | 52.6 ± 3.2 | 60.6 | 5.5 | 52.6 |
| IRM | 56.7 ± 3.3 | 56.6 ± 2.8 | 51.6 ± 0.7 | 51.7 ± 0.7 | 54.2 | 2.5 | 51.6 |
| GroupDRO | 69.7 ± 0.8 | 71.7 ± 0.3 | 72.0 ± 0.2 | 71.4 ± 1.9 | 71.2 | 0.9 | 69.7 |
| MLDG | 60.6 ± 0.3 | 64.6 ± 1.0 | 66.7 ± 0.5 | 55.6 ± 2.4 | 61.9 | 4.2 | 55.6 |
| ARM | 67.5 ± 0.4 | 65.5 ± 1.6 | 66.7 ± 0.6 | 64.7 ± 1.1 | 66.1 | 1.1 | 64.7 |
| VREx | 67.4 ± 1.9 | 70.4 ± 0.1 | 71.2 ± 0.2 | 59.4 ± 4.3 | 67.1 | 4.7 | 59.4 |
| TCRI (cov) | 67.6 ± 0.8 | 64.0 ± 5.4 | 63.0 ± 5.5 | 61.5 ± 4.6 | 64.0 | 2.3 | 61.5 |
| TCRI (HSIC) | 62.2 ± 4.4 | 70.0 ± 1.3 | 67.9 ± 1.4 | 65.4 ± 2.8 | 66.4 | 2.9 | 62.2 |

# E   Theoretical Results

**Lemma E.1.** *(Sufficiency of TCRI for Causal Aggregation). Recall, $X$, $Z_c$, $Z_e$, $Y$ from Figure 1. Let $Z_c, Z_e$ be direct causes and direct effects of $Y$, respectively, and recall that $X$ is a function of $Z_c$ and $Z_e$. If the two representations induced by feature extractors $\Phi, \Psi$ satisfy TCRI, then wlog $I(\Phi(X); Y) \geq I(Z_c; Y)$.*

*Proof.*

   (i) First we define $K$ $Z_c^i$'s to be random variables with non-zero mutual information with $Z_c$ marginally and conditioned on $Y$: $I(Z_c^i; Z_c) > 0$, $I(Z_c^i; Z_c \,|\, Y) > 0$.

  (ii) Furthermore, we have from (i.) that for any pair $Z_c^i, Z_c^j \in \{Z_c^1, \ldots, Z_c^K\}$, $I(Z_c^i; Z_c^j \,|\, Y) \geq 0$, since neither can be made conditionally independent of $Z_c$ given $Y$.

 (iii) Given the total-chain-information criterion, we have that there exist a set of $K$ $Z_c^{i'}s$ across $\Phi(X), \Psi(X)$ s.t. $I(Z_c^1, \ldots, Z_c^K; Y) \geq I(Z_c; Y)$ for some $K$.

 (iv) Combining (ii) and (iii), we have that all $Z_c^{i'}s$ are aggregated in one of the two representations, say $\Phi$, since for any $Z_c^k$ that satisfies (i.), (ii.) $\implies Z_c^k \in \Phi(X)$, and therefore (iii.) $\implies I(\Phi(X); Y) \geq I(Z_c; Y)$.

□

