# OpenReview forum: "Target Conditioned Representation Independence (TCRI); from Domain-Invariant to Domain-General Representations"
_ICLR.cc/2023/Conference — Submitted to ICLR 2023_

### Official Review · Reviewer_ZJFn · 2022-10-22

**Confidence:** 3
**Correctness:** 2
**Technical Novelty And Significance:** 3
**Empirical Novelty And Significance:** 2
**Recommendation:** 5

**Clarity, Quality, Novelty And Reproducibility:**

This paper is moderate in clarity and quality. The proposed method is somewhat novel. Codes are provided to ensure reproducibility.

**Details Of Ethics Concerns:**

No ethics concerns apear.

**Strength And Weaknesses:**

Strength:
- This paper is based on a reasonable SCM to analyze the data-generating process. As a result, the existing method only solves the domain invariant constraints, failing to address the target conditional independence. Hence, TCRI can reasonably achieve better performance than other methods by addressing the target conditional independence.
- This paper has good theoretical support, making TCRI a solid research work.

Weakness:
- It is unclear how the target conditioned representation independence property can be break down into the proposed total-chain-information-criterion $I(\Phi(X), \Psi(X); Y)=I(Z_c, Z_e; Y)$ and $\Phi(X)\indep\Psi(X)|Y \forall e_i$. There is no detailed derivation.
- Furthermore, how the two properties $I(\Phi(X), \Psi(X); Y)=I(Z_c, Z_e; Y)$ and $\Phi(X)\indep\Psi(X)|Y \forall e_i$ can be solved by the proposed $L_{\Phi\oplus\Psi}$ and $L_{\text{CI}}$. Please give more details.
- How to choose the hyper-parameter $\alpha$ and $\beta$ are not discussed. Additionally, there is no experimental analysis of the importance of each loss term.
- The experiments are quite insufficient. All experiments are conducted on coloredMNIST which is a very small dataset synthesized artificially, not a real-world dataset as indicated by the title of Section 6.1. So the realistic applicability is quite doubtful.

**Summary Of The Paper:**

This paper tries to learn domain-general representation by achieving the target conditioned representation independence (TCRI). Specifically, the proposed TCRI method not only addresses the domain invariant property but also assumes that the domain-general feature and domain-specific feature are conditionally independent of each other when given class label $Y$ and domain label $e$. By introducing two constraints to the general loss function which ensure 1) a good predictor, and 2) conditional independence, the TCRI method is demonstrated to achieve domain generality compared to other OOD generalization methods. Theoretical analysis and experimental comparisons are provided to validate TCRI.

**Summary Of The Review:**

I have carefully read the whole paper. This paper is well-motivated, however, there are still some concerns (see weaknesses). If the authors can address my concerns, I will consider raising my score.

---

> ### Author Response · Authors · 2022-11-14
> **Response to Reviewer ZJFn**
>
> Thank you for your time and effort in reviewing this paper and providing valuable constructive feedback. We are happy that you found our work to be well-supported theoretically and consider TCRI to be solid research work. We address your questions below.
>
> * It is unclear how the target conditioned representation independence property can be break down into the proposed total-chain-information-criterion $I(\Phi(X), \Psi(X); Y) = I(Z_c, Z_e; Y)$ and $\Phi(X) \indep \Psi(X) \,|\, Y \forall e_i$. There is no detailed derivation.
>
>   * The two terms you mentioned are part of the definition of the TCRI property, which gives causal aggregation (Lemma 5.3). In addition to the domain-invariance property, this gives domain generalization (Theorem 5.4). We address this question further in the response to the following question.
>
> * Furthermore, how the two properties $I(\Phi(X), \Psi(X); Y) = I(Z_c, Z_e; Y)$ and $\Phi(X) \indep \Psi(X) \,|\, Y \forall e_i$ can be solved by the proposed  $\mathcal{L}_{\Phi \oplus \Psi}$ and $L_{CI}$. Please give more details.
>
>   *   First, we observe that by the assumption of the DAG, $I(X; Y) = I(Z_c, Z_e; Y)$. $L_{\Phi \oplus \Psi}$ (total-chain-information-criterion): This term ensures that the two representations together achieve the optimal within-domain risk. Since we have that both the domain-general and domain-specific contain unique information about the label (and together capture all the information about $Y$ in observed $X$), the model that achieves the optimal within-domain risk will use both types of information, so this term enforces that the representations jointly do the same -- total chain information. $L_{CI}$ is a direct empirical conditional independence penalty for the conditional independence property.
>
>   * Thanks for your feedback; we have updated the paper to make the connections more explicit and clear -- Section 4.
>
> * How to choose the hyper-parameter $\alpha$ and $\beta$ are not discussed. Additionally, there is no experimental analysis of the importance of each loss term.
>
>   *  Perhaps, interestingly, setting $\beta=0$ and $\alpha=1$ recovers IRM as a special case. Other extreme parameter settings do seem meaningful for the problem setting and are explored further. We are completing ablation studies on these hyperparameters and will include them in the final version.
>
> * The experiments are quite insufficient. All experiments are conducted on coloredMNIST which is a very small dataset synthesized artificially, not a real-world dataset as indicated by the title of Section 6.1. So the realistic applicability is quite doubtful.
>
>   * Please see our comment on Real-World Datasets. Additionally, as requested, we have included two additional real-world dataset.

---

### Official Review · Reviewer_N8XZ · 2022-10-24

**Confidence:** 4
**Clarity, Quality, Novelty And Reproducibility:** The paper is hard to read. Mathematic…
**Correctness:** 3
**Technical Novelty And Significance:** 2
**Empirical Novelty And Significance:** 2
**Recommendation:** 3

**Strength And Weaknesses:**

Strengths: A theoretical foundation for generating representations that are domain-invariant and domain generalizable are provided.
Weaknesses: Poorly written paper. Experimental results using synthetic data or variations of MNIST are not convincing.

**Summary Of The Paper:**

This paper provides necessary and sufficient conditions for domain generalization. Experimental results on a few benchmark datasets are provided.

**Summary Of The Review:**

The paper presents an approach for domain generalization. The writing style is harsh. Experimental results are not convincing.

---

> ### Author Response · Authors · 2022-11-14
> **Response to Reviewer N8XZ**
>
> Thank you for your time and effort in reviewing this paper and providing valuable constructive feedback. We are happy that you appreciated the theoretical foundations of our work. We are sorry to hear that you found the paper hard to read and would appreciate and welcome any actionable feedback you may have for us to improve this aspect of the work.

---

### Official Review · Reviewer_FowA · 2022-10-25

**Confidence:** 3
**Correctness:** 3
**Technical Novelty And Significance:** 3
**Empirical Novelty And Significance:** 3
**Recommendation:** 6

**Clarity, Quality, Novelty And Reproducibility:**

The work is mainly clear, but some questions remain to be answered (W2, W3). The proposed method is novel and seems to have good results in the worst-case scenario.

**Strength And Weaknesses:**

**Strong points**

S1. The paper tackles an important problem in domain generalization and proposes a sound method for it. The presented downsides of using representations that are domain-invariant but not necessary domain-general are valid and the proposed method solves them.

S2. By improving on other invariant predictors, like IRM, the proposed method has a large potential for domain generalisation and represents a good contribution.

S3. In general, model selection for domain generalization is a hard problem, and the proposed method seems to achieve a good selection using only the training data.

**Weak Points**:

W1. In practice, even when the proposed model selection is used, the performance for all environments is quite low. This hints that the constraints could be too hard to achieve and the model is hard to optimize. What is the performance of the model if the model selection is using testing domain data (oracle selection)?

W2. Some additional explanations in the proofs are needed.  In step iv of the proof of Lemma 5.3, the variables $Z_c^{i′}$ are aggregated in one of the two representations, but how to choose which one? Is it possible that the variables be aggregated into the representation $\Psi(X)$ that is *not* optimized to be domain-invariant?

W3. Is it necessary to have the two representations? Could an assumption such as
$I(\Phi(X); Y) = I(Z_c, Z_e; Y)$, together with domain-invariance assumption be enough to get the same result as Lemma 5.3 and Theorem 5.4? These assumptions would be satisfied when optimizing $ \mathcal{L_\Phi} + \lambda \mathcal{L}_{IRMv1’}$, i.e. the usual way of training IRM.



**Summary Of The Paper:**

The paper proposes an objective for domain generalization. The main findings involve two types of representations: domain invariant - the distribution of the target conditioned on such a representation is the same across training environments; and domain domain-general - the representation is domain invariant across all possible environments. The paper proves, by construction that a representation that captures only a subset of causal features could be domain-invariant but not necessarily domain-general.

A criterion (Target Conditioned Representation Independence - TCRI ) helpful for domain generality is proposed, using two representations, one domain invariant $\Phi(X)$ and one domain-specific $\Psi(X)$. Informally, this criterion is satisfied when the two representations jointly have enough information while being independent when conditioned on the target. The paper proves that if $\Phi(X)$ and $\Psi(X)$ satisfy TCRI and $\Phi(X)$ satisfies domain independence then $\Phi(X)$ is domain-general.

Then, a loss function is proposed to satisfy this criterion and experimentally it is shown that it can achieve better word-case performance. An important aspect is that the loss used for conditional independence (based on HSIC criterion) can be used for model selection using only training domain data.


**Summary Of The Review:**

The paper was some good observations and it is useful for domain generalization, especially in cases where the worst-case is crucial and model selection is hard.

---

> ### Author Response · Authors · 2022-11-14
> **Response to Reviewer FowA**
>
> Thank you for your time and effort in reviewing this paper and providing valuable constructive feedback. We are happy that you found our observations to be useful and our method to be novel and sound in addressing an important problem in domain generalization. We address your questions below.
>
> * W1: In practice, even when the proposed model selection is used, the performance for all environments is quite low. This hints that the constraints could be too hard to achieve and the model is hard to optimize. What is the performance of the model if the model selection is using testing domain data (oracle selection)?
>
>    * An open question (which we will note in future work) is model selection, as we have demonstrated that observed-domain validation accuracy adds unwanted bias toward what generalizes well within-domain. Our approach is to do a selection based on the assumed DAG properties, but it is known that conditional independence can be hard to measure.
>
>   * Appendix C Table 3 shows results for oracle selection and shows that the constraints do indeed allow for higher accuracy than what is achieved after selection. Additionally, we observe that TCRI still outperforms all other methods in worst-case accuracy, while having comparable (better than most) average accuracy.
>
> * W2: Some additional explanations in the proofs are needed. In step iv of the proof of Lemma 5.3, the variables
>  are aggregated in one of the two representations, but how to choose which one? Is it possible that the variables be aggregated into the representation that is not optimized to be domain-invariant?
>
>   * This is an important point; we have made this clear in the updated draft (Remark 5.6). One limitation of TCRI is a failure mode when the strictly anticausal representation gives a domain invariant predictor. In this case, our constraints allow for the planned domain-general featurizer to be mapped to either domain-general or domain-specific features. However, one of the benefits of having a domain-specific predictor for each observed domain is that we can check if those classifiers are interchangeable. Specifically, if we are in this scenario where the causal features are mapped to the domain-specific feature extractor, we will see that the domain-specific classifiers give similar results when applied to a domain that they were not trained on since they are based on invariant causal mechanisms. This, however, gives a test not a fix for this setting -- we leave a fix for future work.
>
> * W3: Is it necessary to have the two representations? Could an assumption such as $I(\Phi(X); Y) = I(Z_c, Z_e; Y)$, together with domain-invariance assumption be enough to get the same result as Lemma 5.3 and Theorem 5.4? These assumptions would be satisfied when optimizing $L_\Phi + \lambda L_{IRMv1'}$, i.e. the usual way of training IRM.
>
>   * Yes, it is necessary. The assumption $I(\Phi(X); Y) = I(Z_c, Z_e; Y)$ is equivalent to $I(\Phi(X); Y) = I(X; Y)$. This would indeed be sufficient for causal aggregation (Lemma 5.3), however, in the case that there are non-domain-general features that are domain-invariant w.r.t to the observed domains, they would satisfy this constraint and fail to be domain-general. Having two representations (and the total-chain-information-criterion) allows us to enforce the conditional independence property of the DAG to penalize such a case.

---

> > ### Comment · Reviewer_FowA · 2022-11-24
> > **Response to rebuttal**
> >
> >
> > I thank the authors for their rebuttal and ask for some additional clarifications.
> >
> > Regarding point W3. I agree with the authors that  “This [IRM] would indeed be sufficient for causal aggregation (Lemma 5.3), however, in the case that there are non-domain-general features that are domain-invariant w.r.t to the observed domains, they would satisfy this [IRM] constraint and fail to be domain-general.”
> >
> > I would like the authors to expand on the following: “Having two representations (and the total-chain-information-criterion) allows us to enforce the conditional independence property of the DAG to penalize such a case.” How exactly is this case penalized? Do we see this somewhere in the proof?
> >
> > Theorem 5.4 uses only the result of Lemma 5.3 (that IRM also satisfies) and the domain-invariance property (that IRM also satisfies). Thus, it seems like IRM also satisfies Theorem 5.4. Can the authors comment on this?
> >
> > In the second paragraph of the proof, the domain-invariance property fails to constrain the case that the authors mention (non-domain-general features that are domain-invariant w.r.t to the observed domains). So, how is this case penalised by the proposed objective?

---

> > > ### Author Response · Authors · 2022-11-24
> > > **Additional Clarification**
> > >
> > > Thank you for your response and feedback. We are happy to provide more clarification.
> > >
> > > * Regarding point W3. I agree with the authors that “This [IRM] would indeed be sufficient for causal aggregation (Lemma 5.3), however, in the case that there are non-domain-general features that are domain-invariant w.r.t to the observed domains, they would satisfy this [IRM] constraint and fail to be domain-general.”
> > >
> > >   I would like the authors to expand on the following: “Having two representations (and the total-chain-information-criterion) allows us to enforce the conditional independence property of the DAG to penalize such a case.” How exactly is this case penalized? Do we see this somewhere in the proof?
> > >
> > >   Theorem 5.4 uses only the result of Lemma 5.3 (that IRM also satisfies) and the domain-invariance property (that IRM also satisfies). Thus, it seems like IRM also satisfies Theorem 5.4. Can the authors comment on this?
> > >
> > >   * We show that domain invariance property and causal aggregation are sufficient for domain generalization. I believe that the motivating question here is if IRM ($L_\Phi + L_{IRMv1'}$) can implement this.
> > >
> > >   * A point of clarification is that assumption $I(\Phi(X); Y) = I(Z_c, Z_e; Y)$ (which is sufficient for causal aggregation) is not necessarily enforced by IRM. The main issue here is illustrated in Lemma 5.2, which shows that causal subsets can satisfy the domain invariance property for some sets of observed domains (satisfy IRM constraint), but causal subsets are not sufficient for domain generalization (Lemma 5.1). So, IRM does not necessarily satisfy Lemma 5.3.
> > >
> > >   * This necessitates the total-chain-information-criterion -- so that we can account for all domain-general information about $Y$. An important note is that a single representation cannot non-trivially (that is $X$ is already domain-general) satisfy both domain generality and the total-chain-information-criterion -- a domain-general representation should not have spurious information about $Y$ that was in $X$.
> > >
> > >   * To implement the total-chain-information-criterion, a part of the second representation's $\Psi$ objective is to jointly minimize prediction error with the first representation $\Phi$ -- $\Psi$ is not constrained to be domain invariant, any information in $X$ can be used. However, the second part of the $\Psi$ objective is that it is conditionally independent of $\Phi$ given $Y,\, e$. Therefore, we have that all domain-general information is accounted for and, given the conditional independence constraint, is aggregated into $\Phi$, and all of the spurious information is aggregated into $\Psi$. Given this separation, we only need to worry about the case when all spurious information is domain invariant -- we discuss this case below.
> > >
> > > * In the second paragraph of the proof, the domain-invariance property fails to constrain the case that the authors mention (non-domain-general features that are domain-invariant w.r.t to the observed domains). So, how is this case penalised by the proposed objective?
> > >
> > >   * This case is penalized by the conditional independence criterion. We have a separation of the domain-general and spurious information via conditional independence given $Y, e$. Remark 5.6 points out that in the worst case where spurious information is domain invariant, one cannot distinguish between $\Phi$ and $\Psi$. While we do not currently have a fix for this setting, one can identify when one is in this setting by checking if domain-specific predictors on the $\Psi$ are all interchangeable.

---

### Official Review · Reviewer_xK43 · 2022-10-28

**Confidence:** 2
**Clarity, Quality, Novelty And Reproducibility:** See above.
**Correctness:** 3
**Technical Novelty And Significance:** 3
**Empirical Novelty And Significance:** 2
**Recommendation:** 6

**Strength And Weaknesses:**

Strengths:
- This paper is well written.
- This paper gives a new attempt to explain and solve domain generalization in the view of causality.
- This paper contains detailed proof and designs the method based on the analysis.

Weaknesses:
- The method requires a classifier for each domain, and it is evaluated on a three-domain setting {0.1, 0.2, 0.9}. How the results would be if there are more domains? Why choose {0.1, 0.2, 0.9} in the experiment?
- This paper focuses on methodology. Only one real-world dataset is analyzed. It would be better to have more results.
- The method achieves much better results under “-90%” setting, but for the other settings, the results are not as strong as expected, showing the limitation.



**Summary Of The Paper:**

This paper proposed a Target Conditioned Representation Independence objective for domain generalization. It claims that a domain-invariant representation may not extend to test domains and refine it to a domain-general representation. Based on this, this paper proposed the TCRI objective and evaluate it on synthetic and real-world data.

**Summary Of The Review:**

The proposed method is interesting. It lacks evaluation results.

---

> ### Author Response · Authors · 2022-11-14
> **Response to Reviewer xK43**
>
> Thank you for your time and effort in reviewing this paper and providing valuable constructive feedback. We are happy that you found the paper to be well-written and found our method to be novel and interesting. We address your questions below.
>
> * The method requires a classifier for each domain, and it is evaluated on a three-domain setting {0.1, 0.2, 0.9}. How the results would be if there are more domains? Why choose {0.1, 0.2, 0.9} in the experiment?
>
>   * The method achieves much better results under the -90% domain, but for the other domains, the results are not as strong as expected, showing the limitation.
>
>   * The three-domain setting with {+90%, +80%, -90%} is a typical setting that is used as a benchmark, we chose to keep this setting to be consistent with previous work. This setting is especially interesting because of the -90% domain, as the direction of the spurious information (color) and the label is the same between +90% and +80%, however, it is flipped in the -90% domain. So, the spurious information gives the opposite label in the -90% domain that it would give in the +90% and +80% domains. We hypothesize that the reliance on spurious information is the reason we see such a low accuracy in the -90% domain in all of the baselines.
>
>   * We have added more experiments and discussion in Appendix D -- we repeat here. We find that adding an additional domain with +70% yields a small improvement in all of the methods, however, the +90% domain still has significantly lower accuracy for the baselines relative to the other domains while TCRI is stable. When we add a -80% domain, however, all of the methods have a significant increase in the -90\% domain. We hypothesize that this is because the naive empirical risk minimizer for source domains {+90%, +80%, -80%} does not rely on spurious information, since the spurious information (color-label correlation) does not generalize across the three observed source domains, thus the domain-invariant model is closer to domain-general. In conclusion, additional datasets can help average accuracy, and in some cases, worst-case accuracy. The latter, however, seems to only be true for specific types of added domains and is not a worst-case setting that we a concerned about in this work (one where such domains may not be observed).
>
> * The method achieves much better results under -90% setting, but for the other settings, the results are not as strong as expected, showing the limitation.
>   * Please see our comment on Real-World Datasets. Additionally, as requested, we are working to include an additional real-world dataset in the updated version.
>
> * The method achieves much better results under -90% setting, but for the other settings, the results are not as strong as expected, showing the limitation.
>   * Repeating our response to Reviewer FOWA: An open question (which we will note in future work) is model selection, as we have demonstrated that observed-domain validation accuracy adds unwanted bias toward what generalizes well within-domain. Our approach is to do a selection based on the assumed DAG properties, but it is known that conditional independence can be hard to measure.
>
>    * Appendix C Table 3 shows results for oracle selection and shows that the constraints do indeed allow for higher accuracy than what is achieved after selection. Additionally, we observe that TCRI still outperforms all other methods in worst-case accuracy, while having comparable (better than most) average accuracy.

---

### Author Response · Authors · 2022-11-14
**Real-World Dataset**

We thank all of the reviewers for their time and great feedback on our work. As requested, we have included two additional datasets which exhibit the worst-case behavior we aim to be robust against, namely Worst-Case--PACS and Worst-Case--VLCS, as described below. We find that, indeed, our method outperforms the baselines in worst-case accuracy and stability across domains, while still achieving the best average accuracy in the Worst-Case--PACS dataset and maintaining competitive average accuracy in the Worst-Case--VLCS dataset. Additionally, we include the oracle selection accuracies (model selection on held-out target examples) to further show that the regularization from our method is compatible with learning a good predictor. Results below.

We would like to point out that the strength of our method is something of a double edge sword as it removes spurious information, even if this spurious signal happens to generalize to the target domain of interest (discussion in section 6.2). In other words, our method is designed for the case where worst-case accuracy is vital (and one only knows the setting is causal/anticausal). This means that in datasets where the spurious (anticausal) information indeed generalizes, we expect our method will not outperform baselines that use spurious information, since our method loses out on the gains of preserving it.

We suspect that this may be the case in the common baseline datasets -- this hypothesis is supported by observing that ERM (naive) has similar performance as the baseline algorithms across the typical datasets (Gulrajani and Lopez-Paz, 2020). The semi-synthetic Colored MNIST dataset is unique in this respect, as the spurious (anticausal) information does not generalize in this dataset. Specifically, the spurious signal is positively correlated to the target in some domains, and negatively correlated with the target in others.

---

> ### Author Response · Authors · 2022-11-24
> **Worst-Case--VLCS**
>
> **Data Description:** $X$: images, $Y$: animate (bird, dog, person) vs. inanimate (car, chair).
>
> **Domains:** \{\{SUN09, LabelMe\}, \{LabelMe, SUN09\}, \{VOC2007\}\} (Fang et al., 2013). The VOC2007 is the same as in the original dataset. In the \{SUN09, LabelMe\} domain, animate samples are selected from the SUN09 domain while inanimate samples are selected from the LabelMe domain. Conversely, in the \{LabelMe, SUN09\} domain, animate samples are selected from the LabelMe domain, while inanimate samples are selected from the SUN09 domains. The correlation between spurious domain-specific information and the label is flipped between the first two domains and therefore will not generalize.
>
> Worst-Case--VLCS. Domain accuracies and all-domain statistics. TCRI model selection is via the lowest conditional independence score (as in the objective term) on a held-out source domains set. Results highlight that the proposed TCRI achieves the best worst-case domain performance.
>
>                      SxL          LxS            V         Avg    Std.    Min.
>       -------------------------------------------------------------------------
>       ERM.       37.9 (2.8).   47.1 (0.8)    82.7 (1.7)    55.9   19.3    37.9
>
>       IRM.       41.3 (1.6)    46.2 (1.2).   68.6 (2.6)    52.1   11.9    41.3
>
>       GroupDRO.  37.6 (1.1)    50.9 (1.6).   84.2 (0.6)   *57.6*  19.6    37.6
>
>       MLDG.      35.9 (1.3)    48.7 (1.3).   81.7 (1.9).   55.4.  19.3.   35.9
>
>       ARM.       33.4 (1.1)    44.7 (0.8).   83.6 (2.5)    53.9   21.5    33.4
>
>       VREx.      38.1 (2.5)    43.5 (0.8).   80.0 (0.6)    53.9   18.6    38.1
>       -------------------------------------------------------------------------
>       TCRI-COV    49.6 (0.4)   50.4 (0.1)    55.7 (9.0)    51.9   *2.7*  *49.6*
>
>       TCRI-HSIC   49.5 (0.3)   43.9 (3.9)    60.6 (2.7)    51.3    6.9    43.9
>
>
> Worst-Case--VLCS. (Oracle) Domain accuracies -- model selection via held-out accuracy on target domain set (i.e., unobservable in our problem setting). Oracle setting further highlights the soundness of the regularization approach, showing that it does not reject good models.
>
>            SxL    LxS      V
>     ---------------------------------
>     ERM        41.1    50.8   82.9
>
>     IRM        49.6    53.2   65.5
>
>     GroupDRO   45.3    52.5   78.1
>
>     MLDG       40.2    49.2   75.6
>
>     ARM        42.3    47.4   80.5
>
>     VREx       47.1    51.6   78.3
>     ---------------------------------
>     TCRI-COV   49.6    50.4   73.9
>
>     TCRI-HSIC  54.5    52.9   79.0

---

> ### Author Response · Authors · 2022-11-24
> **Worst-Case--PACS**
>
> **Data description:** $X$: images, $Y$: non-urban (elephant, giraffe, horse) vs. urban (dog, guitar, house, person).
>
> **Domains:** \{\{cartoon,  art painting\}, \{art painting,  cartoon\}, \{photo\}\} (Li et al., 2017). The photo domain is the same as in the original dataset. In the \{cartoon, art painting\} domain, urban examples are selected from the original cartoon domain while non-urban examples are selected from the original art painting domain. In the \{art painting, cartoon\} domain, urban examples are selected from the original art painting domain while non-urban examples are selected from the original cartoon domain. Here, the model may use spurious correlations (domain-related information) to predict the labels, however, since these relationships are flipped between domains \{\{cartoon,  art painting\} and \{art painting,  cartoon\}, these predictions will be wrong when generalized to other domains.
>
> Worst-Case--PACS. Domain accuracies and all-domain statistics.TCRI model selection is via the lowest conditional independence score (as in the objective term) on a held-out source domains set. Results highlight that the proposed TCRI achieves the best worst-case domain performance.
>
>                     CxA            AxC          P          Avg       Std        Min
>     ---------------------------------------------------------------------------------
>     ERM          31.2 (1.3)    42.8 (0.7)   97.6 (0.2)     57.2      29.0       31.2
>
>     IRM          30.3 (0.3)    41.4 (1.7)   94.9 (1.4)     55.5      28.2       30.3
>
>     GroupDRO     37.7 (0.7)    42.1 (1.6)   95.7 (0.5)     58.5      26.4       37.7
>
>     MLDG         34.9 (2.4)    41.7 (2.2)   96.8 (0.3)     57.8      27.7       34.9
>
>     ARM          34.1 (0.8)    43.8 (1.1)   96.5 (0.5)     58.1      27.4       34.1
>
>     VREx         37.5 (1.1)    43.0 (0.5)   95.7 (1.5)     58.8      26.2       37.5
>     ---------------------------------------------------------------------------------
>     TCRI-COV     62.8 (0.1)    62.3 (0.2)   65.0 (0.4)    *63.4*    *1.2*      *62.3*
>
>     TCRI-HSIC    35.1 (2.0)    52.5 (4.5)   68.5 (11.5)    52.0      13.7       35.1
>
> Worst-Case--PACS. (Oracle) Domain accuracies -- model selection via held-out accuracy on target domain set (i.e., unobservable in our problem setting). Oracle setting further highlights the soundness of the regularization approach, showing that it does not reject good models.
>
>                   CxA      AxC.      P
>     --------------------------------------
>     ERM           38.4     43.4     95.9
>
>     IRM           62.8     53.9     85.8
>
>     GroupDRO      39.6     49.7     95.7
>
>     MLDG          38.5     40.9     96.2
>
>     ARM           44.2     45.5     94.3
>
>     VREx          55.8     38.7     93.8
>     ------------------------------------------
>     TCRI          62.8     62.3     65.0
>
>     TCRI_HSIC     64.0     62.3     82.4

---

### Decision · Program_Chairs · 2023-01-20

**Decision:**

Reject

**Justification For Why Not Higher Score:**

The score could have been increased with a better an in-depth experimental analyses including
ablation studies on the hyperparameter, applying the method to all the datasets in DomainBed,
a better understanding of the compromise between fullfilling constraints and performance)


**Justification For Why Not Lower Score:**

N/A

**Metareview: Summary, Strengths And Weaknesses:**

The paper introduces an new objective function for domain generalization. It seeks at learning
a domain-general representation by optimizing the so-called target conditioned representation independence
criterion. The paper proposes a theoretical analysis of the TCRI condition.

While some reviewers are positive about the paper, most of them also pointed out that the experimental
analysis is too weak to make the paper passes the bar of acceptance. At this point, suggestions for
improvements mostly are towards these points (more datasets, for instance the DomainBed benchmark,
further experimental ablation studies,...)